# Ablation of STAT3 in Purkinje cells reorganizes cerebellar synaptic plasticity in long-term fear memory network

Jeong-Kyu Han[1,2,3,4†], Sun-Ho Kwon[3,5,6†], Yong Gyu Kim[1,6], Jaeyong Choi[6,7], Jong-Il Kim[6,7], Yong-Seok Lee[1,4,6], Sang-Kyu Ye[1,5,6]*, Sang Jeong Kim[2,3,4,6]*

[1]Department of Physiology, Seoul National University College of Medicine, Seoul, Republic of Korea; [2]Department of Brain and Cognitive Sciences, Seoul National University Graduate School, Seoul, Republic of Korea; [3]Memory Network Medical Research Center, Seoul National University College of Medicine, Seoul, Republic of Korea; [4]Neuroscience Research Institute, Seoul National University College of Medicine, Seoul, Republic of Korea; [5]Department of Pharmacology, Seoul National University College of Medicine, Seoul, Republic of Korea; [6]Department of Biomedical Sciences, Seoul National University College of Medicine, Seoul, Republic of Korea; [7]Department of Biochemistry and Molecular Biology, Seoul National University College of Medicine, Seoul, Republic of Korea

*For correspondence:
sangkyu@snu.ac.kr (S-KY);
sangjkim@snu.ac.kr (SJK)

[†]These authors contributed equally to this work

Competing interests: The authors declare that no competing interests exist.

**Abstract** Emotional memory processing engages a large neuronal network of brain regions including the cerebellum. However, the molecular and cellular mechanisms of the cerebellar cortex modulating the fear memory network are unclear. Here, we illustrate that synaptic signaling in cerebellar Purkinje cells (PCs) *via* STAT3 regulates long-term fear memory. Transcriptome analyses revealed that PC-specific STAT3 knockout (STAT3[PKO]) results in transcriptional changes that lead to an increase in the expression of glutamate receptors. The amplitude of AMPA receptor-mediated excitatory postsynaptic currents at parallel fiber (PF) to PC synapses was larger in STAT3[PKO] mice than in wild-type (WT) littermates. Fear conditioning induced long-term depression of PF–PC synapses in STAT3[PKO] mice while the same manipulation induced long-term potentiation in WT littermates. STAT3[PKO] mice showed an aberrantly enhanced long-term fear memory. Neuronal activity in fear-related regions increased in fear-conditioned STAT3[PKO] mice. Our data suggest that STAT3-dependent molecular regulation in PCs is indispensable for proper expression of fear memory.

## Introduction

Emotional memory processing, such as fear memory, has long been known to engage a large network of brain regions including the cerebellum (*Wager et al., 2015*; *Tovote et al., 2015*). Neuroimaging studies have identified significant neural activity in the midline cerebellum during mental recall of emotional episodes (*Damasio et al., 2000*). The meta-analysis of patients with posttraumatic stress disorders (PTSD) has implicated the cerebellum as one of the activated regions during recall of traumatic memories (*Wang et al., 2016*). In rodents, the inactivation of the cerebellar vermis with tetrodotoxin disrupted fear memory consolidation (*Sacchetti et al., 2002*). Recent research argued that the cerebellum may be involved in the processing of aversive predictions and prediction errors, which has to be added to the neural network underlying emotional domain (*Ernst et al., 2019*).

The cerebellum integrates multisensory information *via* Purkinje cells (PCs), the sole output of cerebellar cortex, and transfers the information to the cerebral cortex, brain stem, basal ganglia, and spinal cords (*Adamaszek et al., 2017*; *Reeber et al., 2013*; *Frey et al., 2011*). Parallel fibers (PFs)

to PC synapses in the vermis lobule V/VI have been reported as a primary site for fear memory formation in the cerebellar cortex (*Sacchetti et al., 2004*). Auditory fear conditioning increased synaptic strength at PFs to PC synapses in the vermis lobule V/VI, and electrically induced long-term potentiation (LTP) in ex vivo slices was occluded after fear conditioning (*Zhu et al., 2007*). It has also been known that GABAergic activity increases at molecular layer interneuron to PC synapses after fear conditioning (*Scelfo et al., 2008*). These studies suggest that the balance of inhibitory-excitatory (I/E) synapses may be necessary for fear memory formation. However, the molecular and cellular mechanisms of the cerebellar cortex modulating the fear memory network are largely unknown.

Here, we focused on the role of signal transducer and activator of transcription 3 (STAT3) in the cerebellum, in order to test the hypothesis that STAT3 in PCs might regulate fear memory network. STAT3 has recently been known to be involved in regulating synaptic plasticity. For example, STAT3 regulates long-term depression (LTD) dependent on N-methyl-D-aspartate receptors (NMDA-Rs) in the hippocampus (*Nicolas et al., 2012*). Genomics research supported the involvement of JAK (Janus kinase)/STAT signaling pathway in the synthesis of the brain's major ion channels and neurotransmitters (*Hixson et al., 2019*). JAK/STAT signaling is also involved in LTD at hippocampal temporoammonic-CA1 synapses (*McGregor et al., 2017*). At the inhibitory synapses, the JAKs can regulate the expression or function of γ-amino-butyric acid receptors (GABA-Rs) (*Lund et al., 2008*; *Raible et al., 2015*). Furthermore, activation of interleukin-6 receptors which is an upstream regulator of STAT3 in the amygdala impaired auditory fear learning (*Hao et al., 2014*).

In this study, we investigated the impacts of specific deletion of STAT3 in PCs on the cerebellar synaptic plasticity and long-term fear memory network. We generated a mutant mouse in which STAT3 was selectively deleted in PCs (STAT3[PKO]) and found that STAT3[PKO] mice showed enhanced α-amino-3-hydroxy-5-methyl-4-isoxazolepropionic acid type glutamate receptor (AMPA-R) expressions, and impaired LTP at the PF–PC synapses. Lastly, STAT3[PKO] mice showed aberrantly enhanced long-term fear memory after fear conditioning, indicating that the deletion of STAT3 in PCs might have implication for PTSD research in future.

## Results

### Generation of PC-specific STAT3 knockout mice

To examine whether cerebellar STAT3 is critically involved in cerebellar synaptic plasticity and fear memory, we generated a PC-specific STAT3 knockout (STAT3[PKO]) mice by crossing STAT3 floxed mice and Pcp2-Cre mice (*Figure 1—figure supplement 1A,B*). First, we verified whether STAT3 protein was exclusively deleted in PCs of STAT3[PKO] mice. In the immunostaining assays, STAT3 expression in PCs was completely depleted in the lobule V/VI of STAT3[PKO] mice (*Figure 1A,B*). We also confirmed that STAT3[PKO] mice showed comparable expression levels of STAT3 to wild-type (WT) mice in other brain regions critical for fear memory such as the amygdala (BLA and CeA), hippocampus (CA3), and paraventricular nucleus of the thalamus (*Figure 1—figure supplement 1C,D*; *Sanders et al., 2003*; *Ressler and Maren, 2019*; *Penzo et al., 2015*). Since STAT3 has a wide range of functions such as neuronal cell growth, survival, and differentiation (*Levy and Darnell, 2002a*; *Di Domenico et al., 2010*; *Yadav et al., 2005*), we further examined whether the ablation of STAT3 impaired cerebellar development. In vitro live cell structural imaging with a calcium indicator revealed that the gross morphology of PCs in mutant mice was similar to that in WT littermates (*Figure 1C*). The quantitative analysis of cerebellar dendritic arborization indicated that deleting STAT3 did not affect dendritic branching processes (*Figure 1D,E*). Furthermore, the number of dendritic spines showed no significant difference between WT and STAT3[PKO] mice (*Figure 1—figure supplement 1E*).

### Altered expressions of synaptic plasticity genes in the STAT3[PKO] mice

Since STAT3 is well known as a multi-functional transcription factor (*Levy and Lee, 2002b*), we conducted RNA-seq analyses to assess its impact on transcription in PCs. By using laser capture microdissection technique, we isolated PC somas from both WT and STAT3[PKO] mice, followed by RNA-seq (*Figure 2A*). The principal component analysis showed that the samples were distinctly separated into WT and STAT3[PKO] groups (*Figure 2—figure supplement 1A*). Differentially expressed gene (DEG) analysis indicated that among 34541 genes, STAT3 depletion upregulated 1987 genes

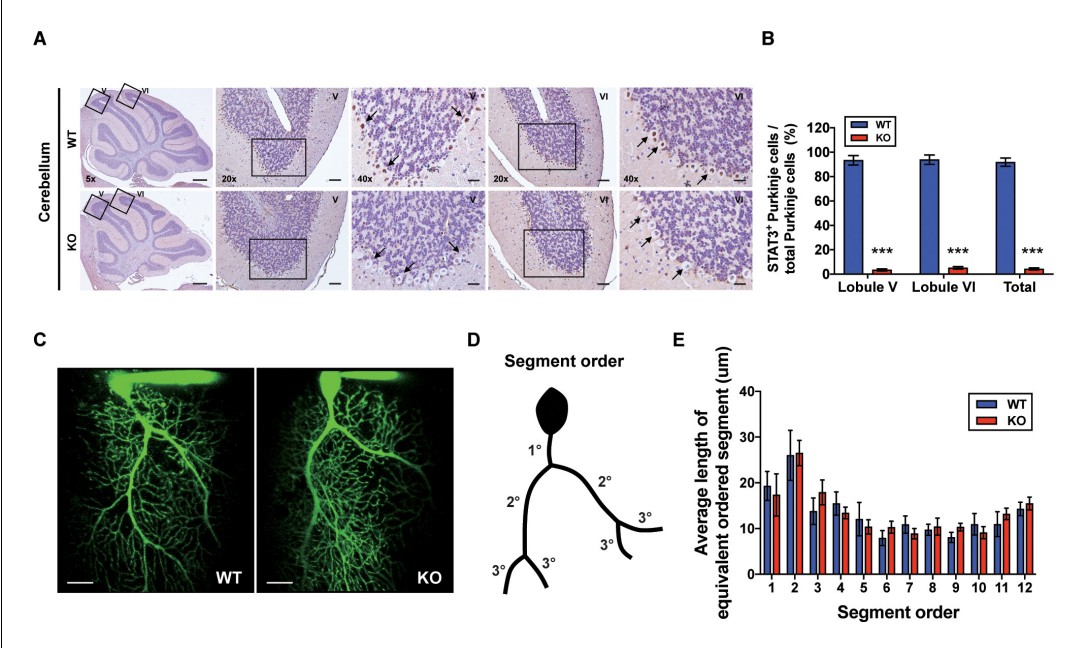

**Figure 1.** Generation of Purkinje cell (PC)-specific STAT3 knockout mice. (A) Immunohistochemistry analysis for STAT3 in cerebellar slices from wild-type (WT) and STAT3$^{PKO}$ mice. Scale bars of 5× image = 1 mm, 20× image = 100 μm, and 40× image = 50 μm. Arrows indicate PC expressing (brown) or not (blue) STAT3. (B) Bar graph shows quantification of STAT3 expression in PCs (WT vs. STAT3$^{PKO}$; Lobule V: 93.3 ± 1.35 vs. 3.43 ± 0.882, Lobule VI: 93.9 ± 1.36 vs. 5.18 ± 1.01, Total: 91.8 ± 1.20 vs. 2.40 ± 0.849, p<0.001, n = 8 slices of five mice per experimental group; two-tailed Student's *t*-test). (C) Two-photon microscopy images of PCs in WT and STAT3$^{PKO}$ mice. Scale bar = 20 μm. (D) Schematic organization of segment order in PCs. (E) Bar graph for the average length of equivalent ordered segment (WT and STAT3$^{PKO}$ groups; n = 3 neurons; two-tailed Student's *t*-test). Data are presented as mean ± SEM.

The online version of this article includes the following figure supplement(s) for figure 1:

**Figure supplement 1.** Generation of STAT3$^{PKO}$ mice model.

(5%) and downregulated 5039 genes (15%) in STAT3$^{PKO}$ mice compared to WT (*Figure 2—figure supplement 1B*). Gene ontology (GO) analysis showed the specific enrichment of upregulated genes in neuronal functions, such as synaptic signaling and trans-synaptic signaling, and the downregulated genes were enriched in GO terms related to G-protein receptor signaling and system processes (*Figure 2B*). Interestingly, AMPA glutamate receptor complex was high ranked in GO analysis containing *Gria1*. By performing an unsupervised hierarchical clustering analysis on synaptic signaling, we also confirmed that each sample was well classified in either WT or STAT3$^{PKO}$ group (*Figure 2C*). Kyoto Encyclopedia of Genes and Genomes (KEGG) pathway has been applied to RNA-seq differential expression analyses to be more compatible with the reality of biological pathways. The KEGG pathway analysis showed that the high ranked upregulated genes are related to retrograde endocannabinoid signaling, synaptic vesicle cycle, dopaminergic synapse, LTD, GABAergic synapses, LTP, and glutamatergic synapses (*Supplementary file 1*). In sum, the transcriptome analyses suggest that deleting STAT3 in PCs leads to differential expression of genes that regulate synaptic plasticity.

## Altered synaptic transmission in the STAT3$^{PKO}$ mice

To examine the physiological impact of STAT3 deletion in PCs on the cerebellar cortex, both excitatory and inhibitory synaptic transmission in PCs of the STAT3$^{PKO}$ mice were examined by whole-cell voltage-clamp recording. We measured AMPA-receptor-mediated miniature excitatory postsynaptic currents (mEPSCs) at PF to PC synapses of the PCs (lobule V/VI) in both WT and STAT3$^{PKO}$ mice (*Figure 2D*). We found that the amplitude of mEPSCs was significantly larger in STAT3$^{PKO}$ mice than in WT, while the frequency of mEPSCs remained unchanged (*Figure 2E,F*). Then, we recorded miniature inhibitory postsynaptic currents (mIPSCs) at molecular layer interneuron (stellate/basket cells) to PC synapses (*Figure 2G*). Unlike the excitatory synaptic transmission, the frequency of mIPSCs was significantly smaller in STAT3$^{PKO}$ mice than in WT, while the amplitude of mIPSCs remained

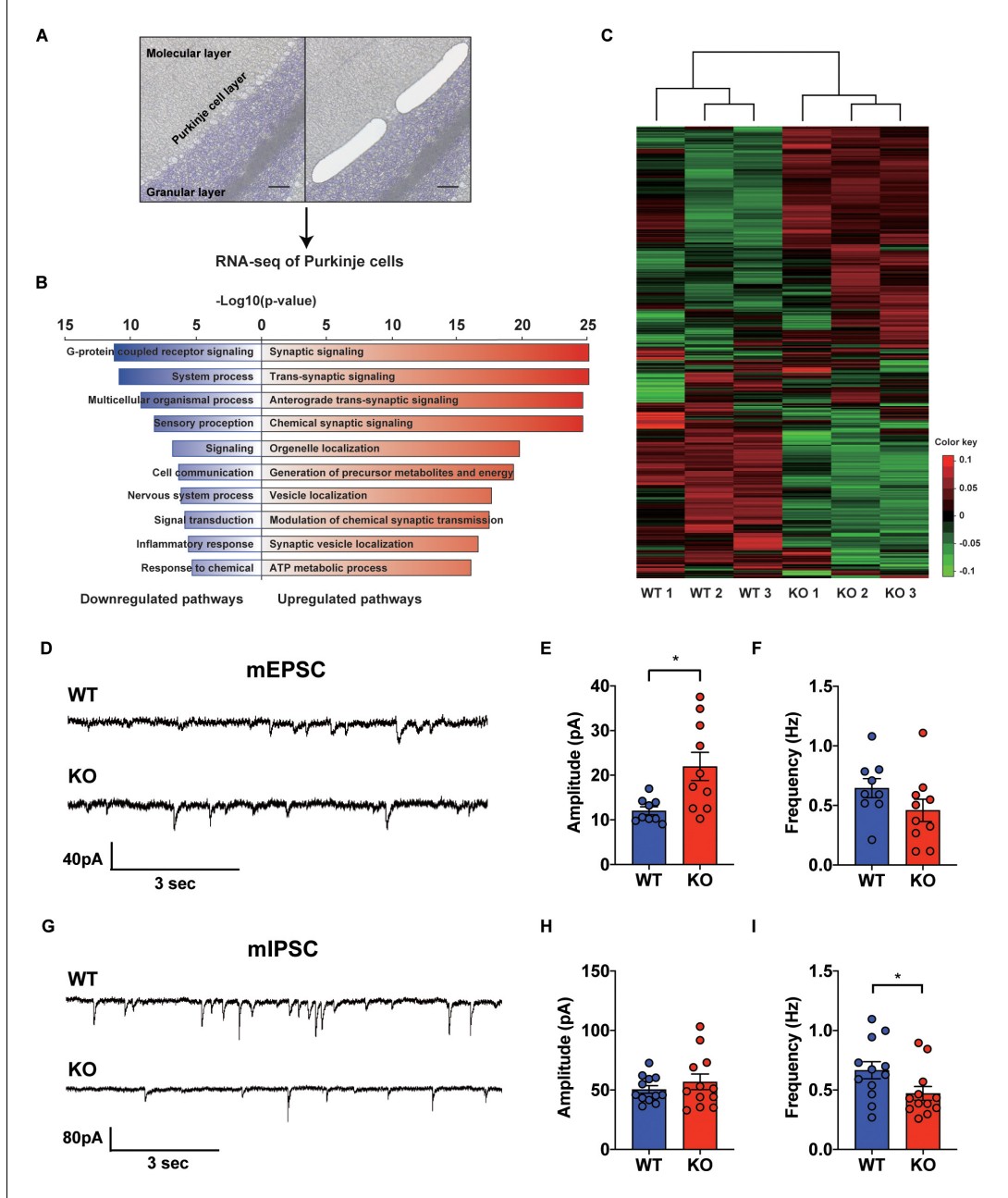

**Figure 2.** Transcriptional regulation of synaptic signaling and transmission in the STAT3[PKO] mice. (A) Schematic representation of RNA sequencing analysis of Purkinje cells (PCs) from wild-type (WT) and STAT3[PKO] mice. PC layers were isolated using laser capture microdissection. Scale bar = 100 μm. (B) The upregulated/downregulated pathways for the STAT3[PKO] mice compared to WT. Gene ontology (GO) analysis was performed using significantly up- and downregulated genes in *Figure 2—figure supplement 1*. Bar graph for the most significant GO pathways. (C) Heatmap depicting 579 differentially expressed transcripts in synaptic signaling. Estimated read count was rlog transformed using DEseq2, then gene centered and normalized for heatmap value. (D) Representative traces for the miniature excitatory postsynaptic currents (mEPSCs) in WT and STAT3[PKO] mice. (E) Bar graph for the average of mEPSCs amplitude (WT vs. STAT3[PKO]; 12.0 ± 0.886 vs. 21.9 ± 3.13, p=0.00980, n = 9, 10 cells; two-tailed Student's *t*-test). (F) Bar graph for the average of mEPSCs frequency (WT vs. STAT3[PKO]; 0.645 ± 0.0803 vs. 0.459 ± 0.0933, p=0.153, n = 9, 10 cells; two-tailed Student's *t*-test). (G) Representative traces for the miniature inhibitory postsynaptic currents (mIPSCs) in WT and STAT3[PKO] mice. (H) Bar graph for the average of mIPSC amplitude (WT vs. STAT3[PKO]; 50.4 ± 3.22 vs. 59.3 ± 7.39, p=0.222, n = 12, 7 cells; two-tailed Student's *t*-test). (I) Bar graph for the mIPSC frequency (WT vs. STAT3[PKO]; 0.666 ± 0.0723 vs. 0.424 ± 0.0316, p=0.0252, n = 12, 7 cells; two-tailed Student's *t*-test). Data are presented as mean ± SEM, and *p<0.05, **p<0.01.

The online version of this article includes the following source data and figure supplement(s) for figure 2:

**Source data 1.** Source data file for di fferentially expressed gene (DEG) analysis.

*Figure 2 continued on next page*

*Figure 2 continued*

**Source data 2.** Source data file for up/down gene pathways.
**Source data 3.** Source data file for gProfiler results (up-genes).
**Source data 4.** Source data file for gProfiler results (down-genes).
**Source data 5.** Source data file for differentially expressed gene (DEG) analysis from DEseq2.
**Figure supplement 1.** Transcriptome analysis of Purkinje cells in wild-type (WT) and STAT3[PKO] mice.

unchanged (*Figure 2H,I*). These electrophysiological data suggest that the altered synaptic strength following the depletion of STAT3 might be attributed to the transcriptional regulation of synaptic plasticity-related genes.

## Increased AMPA receptor expression and occluded LTP in the STAT3[PKO] mice

To specify the direct impacts of synaptic plasticity-related genes, we hypothesized that STAT3 is tightly associated with AMPA glutamate receptor complex. As shown in our GO analysis containing *Gria1*, AMPA glutamate receptor complex was high ranked. Previous studies showed that Hes Family BHLH transcription factor 1 (HES1) and repressor element 1-silencing transcription factor (REST) function as transcriptional repressors of AMPA-type glutamate receptor subunits 1 and 2 (GluA1 and GluA2), respectively (*Lin and Lee, 2012*; *Noh et al., 2012*). STAT3 is known as a transcription factor for HES1 and REST (*Ma et al., 2010*; *Bedini et al., 2008*). Our GO analysis indicated that *Hes1* and *Rest* expressions were reduced in STAT3[PKO] mice, while *Gria1* and *Gria2* were increased (*Figure 3A*). To validate the RNA-seq analysis, we quantified mRNA levels of *Stat3*, *Hes1*, *Rest*, *Gria1*, and *Gria2*. The *Hes1* and *Rest* mRNA expressions were significantly decreased in PC somas; however, *Gria1* and *Gria2* expressions were significantly increased (*Figure 3B*). Then, we examined the expression levels of synaptic proteins, such as postsynaptic density 95 (PSD95), GluA1/2, and calcium/calmodulin-dependent protein kinase II (CaMKII) in cerebellar slices from both WT and STAT3[PKO] groups. We found no significant difference in the total amount of these synaptic proteins between WT and STAT3[PKO] (*Figure 3C,D*). Considering the enrichment of synaptic proteins in dendrites of PCs, we selectively isolated the molecular layer of cerebellar slices and examined the expression of the synaptic proteins. We found that GluA1/2 expressions were significantly increased in the molecular layer of STAT3[PKO] mice compared to WT (*Figure 3E,F*). Furthermore, we examined whether inhibiting STAT3 can induce the same changes in GluA1/2 expression. In order to test the long-term effects of STAT3 inhibition, the hippocampal primary culture treated with a STAT3 inhibitor, such as static, was used as a canonical model for neuronal cell biology. In line with the RNA-seq analysis, we found that both protein and mRNA expressions of *Gria1/2* were increased in STAT3-inhibited neurons, while mRNA expressions of *Hes1* and *Rest* were decreased (*Figure 3—figure supplement 1A,B*). All things considered, these data suggest that the depletion of STAT3 enhances the expression of GluA1/2 by inhibiting the expression of HES1 and REST in PCs.

Since we found that the expression of AMPA-Rs was altered in the STAT3[PKO] mice, we hypothesized that synaptic plasticity in PCs might be affected by STAT3 deletion. We recorded the evoked excitatory postsynaptic currents (eEPSCs) at lobule V/VI with different stimulus intensities (5, 8, 10, and 15 pA). A comparison of the input–output curves of WT and KO in the baseline showed that the amplitude of eEPSCs in KO was higher than that of eEPSCs in WT (*Figure 3G*). We then calculated the paired-pulse ratio and observed no changes in either paired-pulse facilitation or depression (*Figure 3H*). Previous studies have shown that fear conditioning induced LTP of the PF to PC synapses and also occluded further LTP in ex vivo slices, suggesting that LTP might be a key cellular mechanism for fear memory formation (*Sacchetti et al., 2004*; *Zhu et al., 2007*). We electrically stimulated the PF at 1 Hz to induce LTP in WT. The AMPA-receptor-mediated EPSCs in WT mice remained potentiated for 50 min (*Figure 3I*). Interestingly, the same stimulation protocol induced LTD-like synaptic plasticity in STAT3[PKO] mice (*Figure 3J*). Previous research has shown that enhanced AMPA-R activity switches LTP to LTD in the cerebellar cortex (*van Beugen et al., 2014*). These results indicate that the deletion of STAT3 may alter the polarity of cerebellar synaptic plasticity.

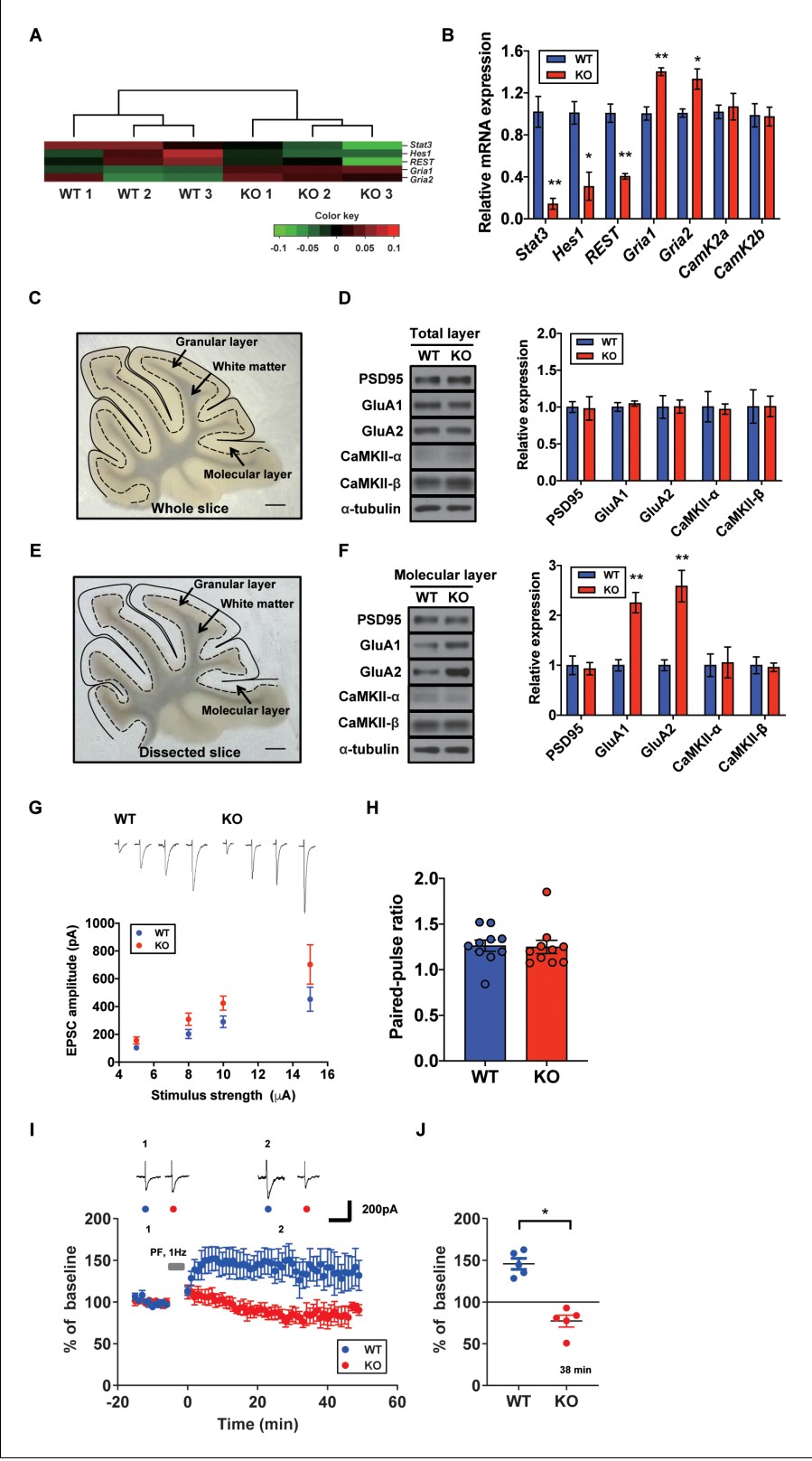

**Figure 3.** Increased AMPA receptor expression and occluded long-term potentiation (LTP) in the STAT3[PKO] mice. (**A**) Heatmap showing normalized transcripts of *Stat3*, *Hes1*, *Rest*, *Gria1*, and *Gria2* in wild-type (WT) and STAT3[PKO] mice. (**B**) Relative mRNA levels of *Stat3*, *Hes1*, *Rest*, *Gria1*, *Gria2*, *CamK2a*, and *CamK2b* in Purkinje cells (PCs) of the cerebellar slices (WT vs. STAT3[PKO]; *Stat3*: 1.02 ± 0.147 vs. 0.144 ± 0.0523, p=0.00490, *Hes1*: 1.01 ± 0.108 vs.

*Figure 3 continued on next page*

*Figure 3 continued*

0.310 ± 0.134, p=0.0151, *Rest*: 1.00 ± 0.0874 vs. 0.406 ± 0.0264, p=0.00270, *Gria1*: 1.00 ± 0.0630 vs. 1.40 ± 0.0373, p=0.00550, *Gria2*:1.00 ± 0.0389 vs. 1.33 ± 0.0968, p=0.0358, *CamK2a*: 1.02 ± 0.0647 vs. 1.07 ± 0.126, p=0.741, and *CamK2b*: 0.988 ± 0.111 vs. 0.975 ± 0.0894, p=0.935, n = 3 mice; two-tailed Student's *t*-test. Relative mRNA levels were normalized to the signals of GAPDH expression. (C) Representative image of the whole cerebellar slice. Scale bar = 1 mm. (D) Western blot analysis of PSD95, GluA1, GluA2, CaMKII-α, CaMKII-β, and α-tubulin in the total layers of cerebellar slice (WT vs. STAT3$^{PKO}$; PSD95: 1.00 ± 0.0726 vs. 0.982 ± 0.159, p=0.918, GluA1: 1.00 ± 0.0594 vs. 1.04 ± 0.0349, p=0.541, GluA2: 1.00 ± 0.154 vs. 1.00 ± 0.0897, p=0.980, CaMKII-α: 1.00 ± 0.206 vs. 0.973 ± 0.0700, p=0.888, and CaMKII-β: 1.00 ± 0.226 vs. 1.01 ± 0.139, p=0.993, n = 3 mice; two-tailed Student's *t*-test). (E) Representative image of the dissected cerebellar slice. Scale bar = 1 mm. (F) Western blot analysis of PSD95, GluA1, GluA2, CaMKII-α, CaMKII-β, and α-tubulin in the molecular layer of cerebellar slice (WT vs. STAT3$^{PKO}$; PSD95: 1.00 ± 0.187 vs. 0.932 ± 0.123, p=0.776, GluA1: 1.00 ± 0.115 vs. 2.25 ± 0.205, p=0.00590, GluA2: 1.00 ± 0.109 vs. 2.58 ± 0.314, p=0.0088, CaMKII-α: 1.00 ± 0.227 vs. 1.05 ± 0.309, p=0.888, and CaMKII-β: 1.00 ± 0.169 vs. 0.960 ± 0.0871, p=0.845, n = 3 mice; two-tailed Student's *t*-test). Quantification of western blot analysis was obtained with relative densitometry and normalized with α-tubulin. (G) Representative traces and graph for evoked excitatory postsynaptic current (eEPSC) amplitudes of PC in WT and STAT3$^{PKO}$ groups (WT and STAT3$^{PKO}$; genotype × stimulus strength interaction: $F_{(3,59)}$=0.8479, p=0.4733; genotype effect: $F_{(1,59)}$=10.89, p=0.0016; stimulus strength effect: $F_{(3,59)}$=17.77, p<0.0001; n = 10, 10 cells; two-way ANOVA with Bonferroni correction). (H) The graph for paired pulse ratio at the 100 ms intervals (WT vs. STAT3$^{PKO}$; p=0.8896, n = 10, 10 cells; two-tailed Student's *t*-test). (I) Representative traces and graph for LTP induction in WT and STAT3$^{PKO}$ groups (WT and STAT3$^{PKO}$ groups, respectively, genotype × time interaction: $F_{(59,560)}$=1.23, p=0.117; genotype effect: $F_{(1,560)}$=261.3, p<0.0001; time effect: $F_{(59,560)}$=0.766, p=0.897; n = 7, 5 cells; two-way ANOVA with Bonferroni correction). All EPSC amplitudes were normalized in percentile. (J) Graph for LTP induced EPSC amplitudes at 38 min (WT vs. STAT3$^{PKO}$; 145 ± 18.7 vs. 81.1 ± 12.2, p=0.0411, n = 7, 5 cells; two-tailed Student's *t*-test). Data are presented as mean ± SEM, and $^*$p<0.05, $^{**}$p<0.01.

The online version of this article includes the following figure supplement(s) for figure 3:

**Figure supplement 1.** Mechanism underlying the STAT3-modulated AMPA receptor expressions in hippocampal neuron model.

## Aberrant long-term fear memory in the STAT3$^{PKO}$ mice

In order to determine whether PC-specific STAT3 is responsible for cognitive/emotional behavior, we subjected the STAT3$^{PKO}$ mice to a series of behavioral tests, including fear memory tests. First, we used the Pavlovian fear conditioning paradigm to examine how STAT3 contributed to fear memory processing. PC-specific STAT3 deletion did not impair the acquisition phase of auditory fear conditioning (*Figure 4A*). In contextual and cued fear conditioning, STAT3$^{PKO}$ mice displayed a comparable level of short-term fear memory to WT mice as assessed by freezing (*Figure 4B*). Interestingly, the long-term fear memory of STAT3$^{PKO}$ mice was significantly enhanced compared to the WT mice only in the auditory cue test (*Figure 4C*). But both WT and STAT3$^{PKO}$ mice did not show significant difference in the contextual memory test (*Figure 4C*). Next, we performed passive avoidance, fear-potentiated startle, and pre-pulse inhibition (PPI) tests (*Figure 4D,E,F,G,H*), and found that the avoidance memory significantly increased in the STAT3$^{PKO}$ group after fear conditioning (*Figure 4D*). The STAT3$^{PKO}$ mice showed comparable startle reflex, including the auditory startle response and PPI, to the WT mice (*Figure 4E,G*). However, 24 hr after fear conditioning, the STAT3$^{PKO}$ mice showed more exaggerated startle responses compared to the WT group (*Figure 4F*). It is possible that STAT3 deletion may have sensitized their perceptual experience of painful stimuli, which ultimately led to improve retention of fear memory and its expression. By monitoring the video recordings, however, we analyzed the number of jumps was not significantly different between WT and STAT3$^{PKO}$ mice (*Figure 4I*). In addition, STAT3$^{PKO}$ mice showed no significant difference in anxiety-like behavior in the elevated plus-maze tests compared to WT (*Figure 4J*). Since the cerebellum is known to be important for motor learning and coordination, we further examined whether the STAT3$^{PKO}$ mice showed any motor deficits. We found no correlation between PC-specific STAT3 expression and motor-related behaviors, including locomotor activity, motor coordination, and vestibulo-ocular reflex (VOR) (*Figure 4K,L,M*). Taken together, the results indicate that the ablation of STAT3 in PCs may be selectively involved in inducing long-term fear memory.

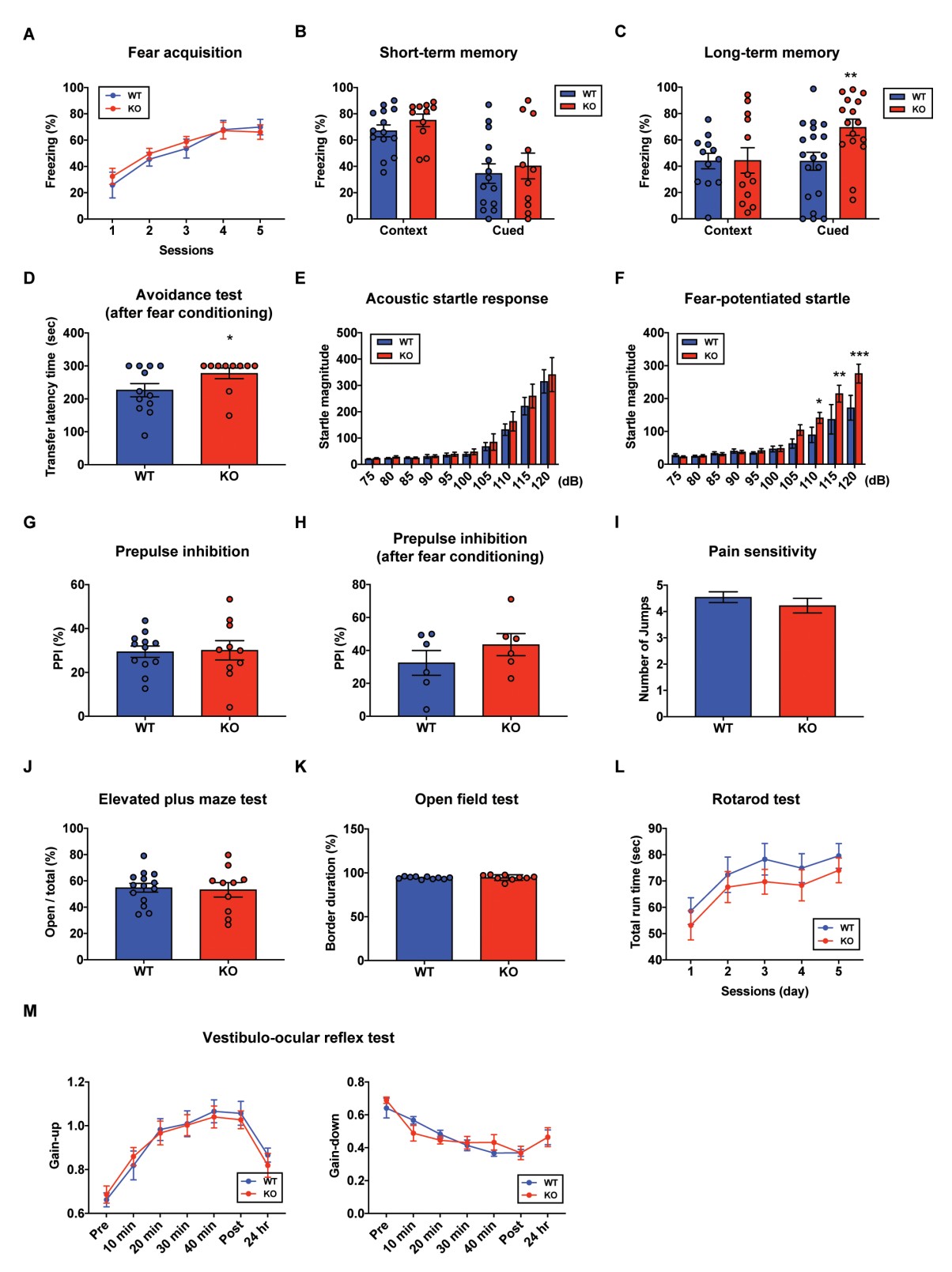

**Figure 4.** Aberrant long-term fear memory in the STAT3$^{PKO}$ mice. (**A**) The percentage of freezing time spent during fear acquisition (genotype × time interaction: $F_{(4,56)}$=0.276, p=0.892; genotype effect: $F_{(1,14)}$=0.260, p=0.617; time effect: $F_{(4,56)}$=14.9, p<0.001, n = 7, nine mice; wild-type (WT) and STAT3$^{PKO}$ groups, respectively; two-way ANOVA with Bonferroni correction). (**B**) The percentage of freezing time spent in short-term contextual and cued fear memory tests (WT vs. STAT3$^{PKO}$, context: 67.0 ± 4.54 vs. 75.0 ± 4.79, p=0.198, n = 14, 11 mice, Mann–Whitney test; WT vs. STAT3$^{PKO}$, cued:
*Figure 4 continued on next page*

*Figure 4 continued*

34.5 ± 7.44 vs. 40.2 ± 9.84, p=0.702, n = 14, 11 mice, Mann–Whitney test). (C) The percentage of freezing time spent in long-term contextual and cued fear memory tests (WT vs. STAT3$^{PKO}$, context: 43.9 ± 5.84 vs. 44.3 ± 9.67, p=0.967, n = 12, 12 mice, two-tailed Student's *t*-test; WT vs. STAT3$^{PKO}$, cued: 43.7 ± 6.80 vs. 69.4 ± 6.21, p=0.00960, n = 19, 16 mice, two-tailed Student's *t*-test). (D) The transfer latency time was assessed in the step-through passive avoidance test (WT vs. STAT3$^{PKO}$, 226 ± 20.1 vs. 277 ± 16.1, p=0.0452, n = 12, 10 mice, Mann–Whitney test). (E) The startle magnitude in a wide range of sound intensities was assessed in the acoustic startle response test (n = 12, 12 mice; WT and STAT3$^{PKO}$ groups, genotype × sound interaction: $F_{(9,198)}$=0.208, p=0.993; genotype effect: $F_{(1,22)}$=0.423, p=0.521; sound effect: $F_{(9,198)}$=48.7, p<0.001; two-way ANOVA with Bonferroni correction). (F) After fear conditioning, the startle magnitude in a wide range of sound intensities was assessed in the acoustic startle response test (n = 10, 10 mice; WT and STAT3$^{PKO}$ groups, genotype × sound interaction: $F_{(9,162)}$=2.84, p=0.00390; genotype effect: $F_{(1,18)}$=4.42, p=0.0496; sound effect: $F_{(9,162)}$=36.3, p<0.001; two-way ANOVA with Bonferroni correction). (G) Comparison between WT and STAT3$^{PKO}$ mice in prepulse inhibition test (WT vs. STAT3$^{PKO}$, 29.42 ± 2.57 vs. 30.0 ± 4.39, p=0.895, n = 12, 10 mice, two-tailed Student's *t*-test). (H) The percentage of prepulse inhibition of WT and STAT3$^{PKO}$ mice after fear conditioning (WT vs. STAT3$^{PKO}$, 32.4 ± 7.51 vs. 43.5 ± 6.71, p=0.297, n = 6, six mice, two-tailed Student's *t*-test). (I) Measurement of the number of jumps during fear learning session for pain sensitivity (WT vs. STAT3$^{PKO}$; p=0.374, n = 11, 10 mice, Mann–Whitney test). (J) The percentage of time spent in the open arms of plus arms (WT vs. STAT3$^{PKO}$, 54.7 ± 3.34 vs. 53.1 ± 5.50, p=0.797, n = 14, 10 mice, two-tailed Student's *t*-test). (K) The percentage of time spent in the border of the open field (WT vs. STAT3$^{PKO}$, 94.3 ± 0.438 vs. 94.6 ± 1.05, p=0.808, n = 10, nine mice, two-tailed Student's *t*-test). (L) Total run time on the rotating drum (n = 14, 15 mice; WT and STAT3$^{PKO}$ groups, respectively, genotype × session interaction: $F_{(4,108)}$=0.0520, p=0.994; genotype effect: $F_{(1,27)}$=0.143, p=0.241; session effect: $F_{(4,108)}$=6.12, p<0.001; two-way repeated measured ANOVA with Bonferroni correction). (M) Vestibulo-ocular reflex (VOR) gain through 50 min of gain-up and -down training sessions, and at 24 hr point after training (gain-up: n = 9, seven mice; WT and STAT3$^{PKO}$ groups, respectively, genotype × time interaction: $F_{(6,84)}$=0.396, p=0.879; genotype effect: $F_{(1,14)}$=0.0247, p=0.877; time effect: $F_{(6,84)}$=31.8, p<0.001, gain-down: n = 6 mice; WT and STAT3$^{PKO}$ groups, genotype × time interaction: $F_{(6,60)}$=0.169, p=0.138; genotype effect: $F_{(1,10)}$=0.00200, p=0.965; time effect: $F_{(6,60)}$=28, p<0.001; two-way ANOVA with Bonferroni correction). Data are presented as mean ± SEM, and *p<0.05, **p<0.01, ***p<0.001.

## Altered learning-induced long-term synaptic plasticity of fear memory in STAT3$^{PKO}$ mice

Since STAT3$^{PKO}$ mice showed the enhancement of long-term fear memory, we tested how enhanced AMPA-Rs in STAT3$^{PKO}$ mice contribute to the long-term fear memory formation. We measured the amplitude of eEPSCs at lobule V/VI with different stimulus intensities (5, 8, 10, and 15 pA). Twenty-hours after fear conditioning, eEPSCs increased at PF to PC synapses of WT mice as previously reported (*Sacchetti et al., 2004*; *Figure 5A*); however, eEPSCs decreased in STAT3$^{PKO}$ mice (*Figure 5B*). We found no changes in either paired-pulse facilitation or depression (*Figure 5C*). In cerebellar fear conditioning, feedforward inhibition has been suggested as a neural circuitry mechanism for timing control (*Sacchetti et al., 2009*; *Heiney et al., 2014*). To test whether STAT3 is involved in the LTP of inhibitory synapses, we recorded mIPSCs at molecular layer interneuron to PC synapses, before and after fear conditioning. PC-specific STAT3 deletion did not play a critical role in fear conditioning-induced potentiation at inhibitory synapses. LTP of inhibitory synapses was observed in both WT and STAT3$^{PKO}$ mice after fear conditioning (*Figure 5D,E,F*). In particular, mIPSC frequency increased in both WT and STAT3$^{PKO}$ groups after fear conditioning, as previously reported (*Scelfo et al., 2008*; *Figure 5E*).

Since deleting STAT3 in PC selectively affects excitatory synaptic plasticity in fear memory microcircuits, we hypothesized that PC-specific STAT3 deletion might modulate the output of PCs. To obtain a physiological assessment of the output control of PCs, we conducted the cell-attached spike recording experiments. The spontaneous firing rate increased in WT mice after fear conditioning, but not in STAT3$^{PKO}$ mice (*Figure 5G,H*). These results indicated that basal neural firing rate would increase in WT mice, but not in STAT3$^{PKO}$ mice before fear memory retrieval.

## Minimal effects on intrinsic excitability of PCs in the STAT3$^{PKO}$ mice

Since PCs are well known to be highly excitable, with high spontaneous firing rates, we tested whether PC-specific STAT3 deletion affects the intrinsic excitability in the presence of NBQX and picrotoxin (PTX). Both the number of spikes and mean firing rate similarly increased in both WT and STAT3$^{PKO}$ groups when PCs were injected a series of increasing current steps at 100 pA intervals (*Figure 5—figure supplement 1A,B,C,D*). The analysis of the membrane properties of PCs showed that the input resistance of STAT3$^{PKO}$ groups was higher than that of WT (*Figure 5—figure supplement 1E,F*). These data indicate that PC-specific STAT3 is not critically involved in the intrinsic excitability of PCs.

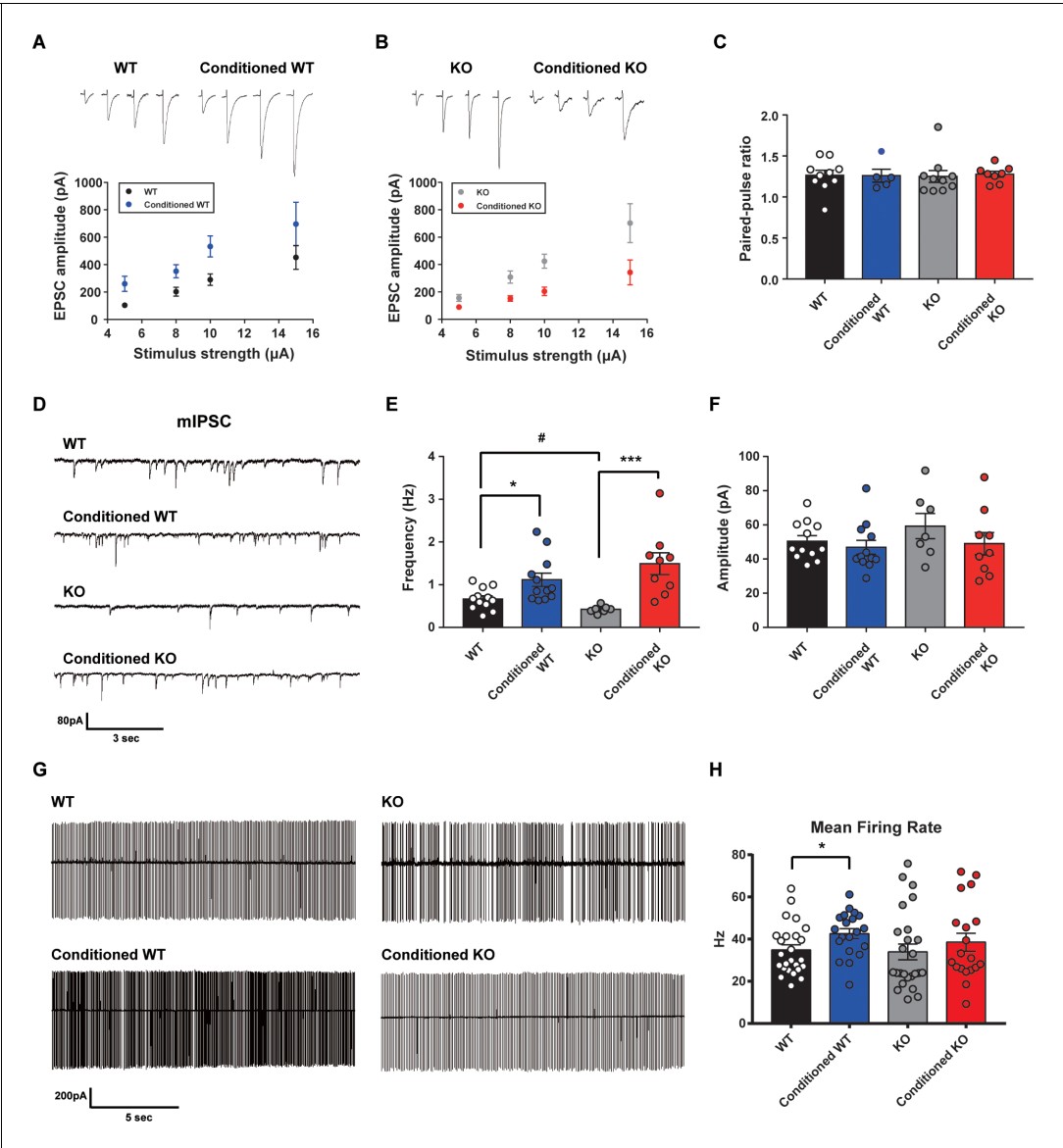

**Figure 5.** Altered learning-induced long-term synaptic plasticity of fear memory in STAT3^PKO mice. (**A**) Representative traces of synaptic strength of wild-type (WT) mice before and after fear conditioning. The plot graph shows the evoked excitatory postsynaptic current (eEPSC) amplitude of WT and fear-conditioned WT mice (n = 10, 8 cells; WT and conditioned WT groups, respectively, group × stimulus strength interaction: $F_{(3,51)}=0.329$, p=0.804; group effect: $F_{(1,51)}=17.4$, p<0.001; stimulus strength effect: $F_{(3,51)}=11.1$, p<0.001; two-way ANOVA with Bonferroni correction). (**B**) Representative traces of synaptic strength of STAT3^PKO mice before and after fear conditioning. The plot graph shows the eEPSC amplitude of STAT3^PKO mice and fear-conditioned STAT3^PKO mice (n = 10, 8 cells; KO and conditioned KO groups, respectively, group × stimulus strength interaction: $F_{(3,55)}=2.38$, p=0.0788; group effect: $F_{(1,55)}=29.4$, p<0.001; stimulus strength effect: $F_{(3,55)}=17.6$, p<0.001; two-way ANOVA with Bonferroni correction). (**C**) The graph for paired pulse ratio at the 100 ms intervals (WT, conditioned WT, KO, and conditioned KO groups, n = 10, 5, 10, and 8 cells; $F_{(3,29)}=0.0301$, p=0.992; one-way ANOVA). (**D**) Representative traces for the miniature inhibitory postsynaptic currents (mIPSCs) of WT and STAT3^PKO mice under control or fear conditions (n = 12, 12, 7, 9 cells; WT, Conditioned WT, KO, and Conditioned KO groups, respectively). (**E**) Bar graph for the mIPSC frequency (WT vs. Conditioned WT, 0.666 ± 0.0723 vs. 1.11 ± 0.155, p=0.0140, Mann–Whitney test; WT vs. KO, 0.666 ± 0.0723 vs. 0.424 ± 0.0316, p=0.0252, two-tailed Student's t-test; KO vs. Conditioned KO, 0.424 ± 0.0316 vs. 1.49 ± 0.254, p=0.00100, Mann–Whitney test). (**F**) Bar graph for the mIPSC amplitude (WT vs. Conditioned WT, 50.4 ± 3.22 vs. 46.8 ± 4.07, p=0.248, Mann–Whitney test; WT vs. KO, 50.4 ± 3.22 vs. 59.3 ± 7.39, p=0.222, two-tailed Student's t-test; KO vs. Conditioned KO, 59.3 ± 7.39 vs. 49.0 ± 6.54, p=0.314, two-tailed Student's t-test). (**G**) Representative traces for tonic pattern firings of Purkinje cells (PCs) in WT and STAT^PKO mice, before and after fear conditioning (n = 25, 20, 25, 20 cells; WT, Conditioned WT, KO, and Conditioned KO groups, respectively). (**H**) Mean firing rate of total patterns of PCs in WT and STAT^PKO mice in naïve and fear-conditioned groups (WT vs. Conditioned WT, 34.8 ± 2.40 vs. 42.6 ± 2.33, p=0.0282, two-tailed Student's t-test; WT vs. KO, 34.8 ± 2.40 vs. 33.9 ± 3.71, p=0.277, Mann–Whitney test; KO vs. Conditioned KO, 33.9 ± 3.71 vs. 38.5 ± 4.23, p=0.142, Mann–Whitney test). Data are presented as mean ± SEM, and *p<0.05, **p<0.01. The online version of this article includes the following figure supplement(s) for figure 5:

*Figure 5 continued on next page*

*Figure 5 continued*

**Figure supplement 1.** Whole cell current-clamp recordings for measuring intrinsic excitability of Purkinje cell (PC) in wild-type (WT) and STAT3$^{PKO}$ mice, (n = 24, 22, 29, 22 cells; WT, Conditioned WT, KO, and Conditioned KO groups, respectively).

## Increased neural activity of fear-related regions in the STAT3$^{PKO}$ mice

Although we examined the underlying mechanism of fear memory in cerebellar cortex, PC-specific STAT3 deletion may have a more widespread impact on the brain's fear memory network in general. Previous studies have shown that removal of only a few highly interconnected areas (high-degree nodes) of the fear network was enough to disrupt fear memory consolidation (*Vetere et al., 2017*; *Silva et al., 2019*). Under this assumption, we performed immunohistochemistry for detecting c-fos expressions in the brain after auditory long-term fear memory retrieval tests. The immediate-early gene c-fos has been represented as a marker for neural activation in memory and psychiatric disorders (*Gallo et al., 2018*). Interestingly, we found that c-fos expressions were significantly increased in several fear-related areas, such as paraventricular nucleus of thalamus, basolateral amygdala, and prelimbic cortex, in the fear-conditioned STAT3$^{PKO}$ mice compared to conditioned WT mice (*Figure 6A,B,C,D,E,F*; *Penzo et al., 2015*; *Silva et al., 2019*). In addition, we confirmed that c-fos expressions in hippocampus were increased in both WT and STAT3$^{PKO}$ mice after tone stimulation. However, we did not find any difference of c-fos expressions between WT and STAT3$^{PKO}$ mice, suggesting that altered cerebellar synaptic plasticity may not affect hippocampal-dependent fear memory (*Figure 6—figure supplement 1A,B*). Together, these results indicate that PC-specific STAT3 may activate a network of regions that mediate long-term fear memory consolidation.

## Discussion

Here, we provide converging evidence that STAT3 plays a critical role in cerebellar synaptic plasticity and long-term fear memory. We found that PC-specific STAT3 controls the transcriptional regulation of AMPA-Rs, cerebellar synaptic plasticity, and consequently long-term memory of fear behavior. Ablation of STAT3 in the cerebellum affected fear memory network in the whole brain.

Our transcriptomic analyses provide the first evidence that STAT3 is involved in regulating synaptic transmission and plasticity through a genomic mechanism (*Figure 2*). We found that the expression of transcriptional repressors, HES1 and REST, decreased in the STAT3$^{PKO}$ mice. Both HES1 and REST constitutively repress GluA1 and GluA2 expressions thus increasing GluA1/2 expression increased in the STAT3$^{PKO}$ groups. A series of changes in the transcriptional regulation in STAT3$^{PKO}$ mice subsequently altered the polarity of synaptic plasticity and fear memory, implicating STAT3 as a gatekeeper for optimal AMPA-R expression. It is noteworthy that negative regulators play a functionally important role in synaptic plasticity and memory formation (*Lee and Silva, 2009*; *Schoch and Abel, 2014*). Suppressing the expression of memory-suppressing genes such as calcineurin was shown to enhance memory (*Baumgärtel et al., 2008*).

Given the above results that entail the STAT3 deletion in PCs, enhanced AMPA-R expression and increased excitatory synaptic transmission may underlie the change in the polarity of synaptic plasticity. The enhanced expression of AMPA-Rs strengthens excitatory synaptic inputs to PCs and depolarizes the membrane potential to activate the voltage-gated calcium channels (VGCCs) (*Ito, 2002*). Opening of VGCCs results in calcium influx, giving rise to an increase in the dendritic calcium concentration, which may lead to LTD induction rather than LTP (*Ito, 2002*). A previous study has shown that the enhanced activity of AMPA-R could cause the polarity change in synaptic plasticity, which accords well with our current data (*Figure 3*; *van Beugen et al., 2014*). Together, stimulation of PF–PC synapses may induce higher calcium concentration in the STAT3$^{PKO}$ mice, resulting in LTD, although this remains to be investigated.

To explain the functional consequence of the change in the polarity of synaptic plasticity, it is worth noting that both excitatory and inhibitory synaptic plasticity are involved in freezing behavior (*Sacchetti et al., 2009*). After fear conditioning, LTP at both excitatory and inhibitory synapses occurred in naïve mice (*Sacchetti et al., 2004*; *Zhu et al., 2007*; *Scelfo et al., 2008*). However, since STAT3$^{PKO}$ mice showed LTD-like plasticity at the excitatory synapses after fear conditioning, the decreased strength of excitatory synapses would affect aberrant freezing behavior. In the case of

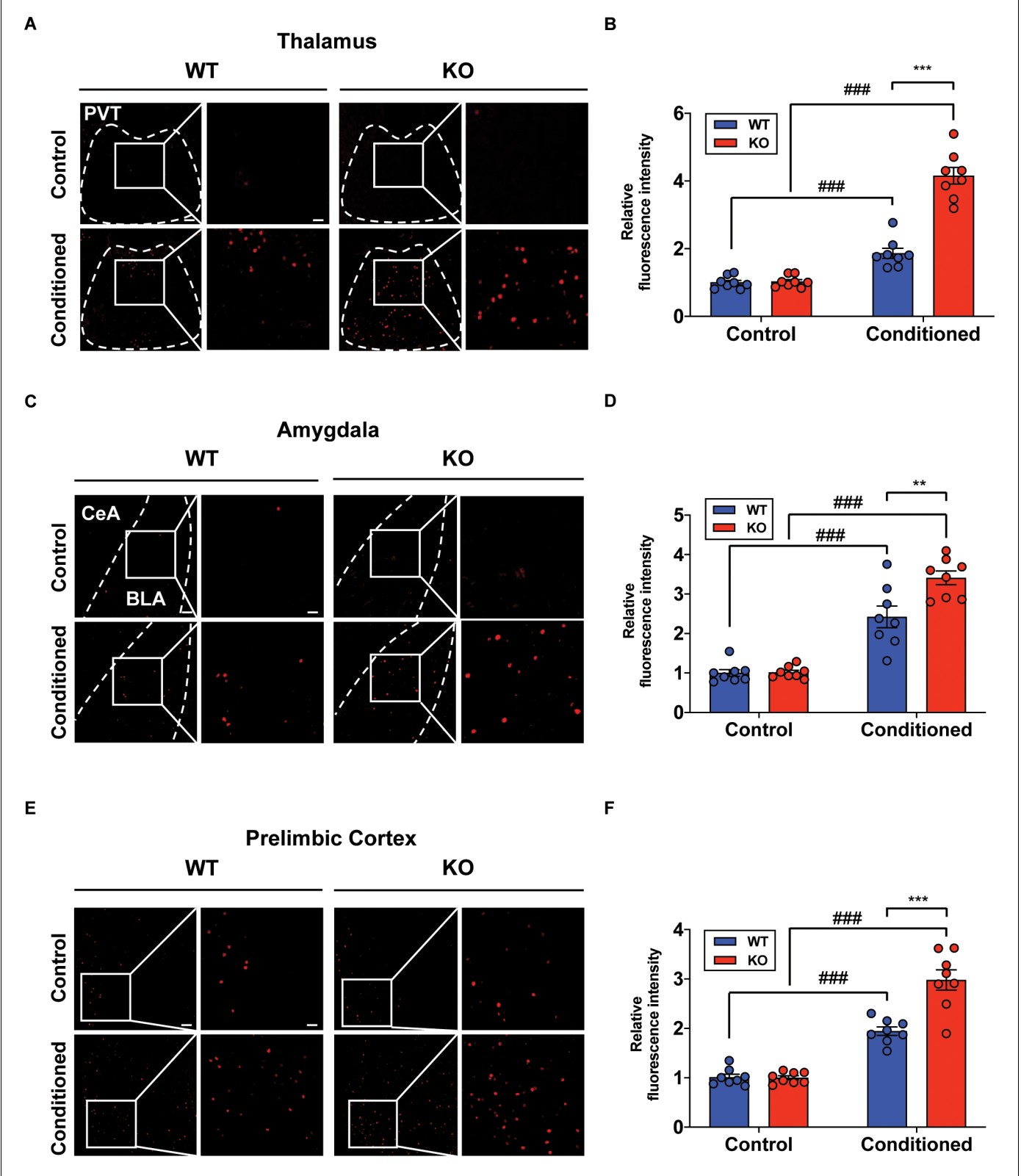

**Figure 6.** Increased neural activity of fear-related regions in the STAT3[PKO] mice. (**A**) Representative image of the c-fos staining on the nucleus of paraventricular thalamus. Scale bar = 250 μm. Scale bar of the enlarged image = 50 μm. (**B**) Quantification of relative fluorescence intensity for c-fos signals, before and after fear conditioning on the paraventricular nucleus of the thalamus (Control; wild-type [WT] and STAT3[PKO], Conditioned; WT and STAT3[PKO], respectively, n = 8 slices of three mice per experimental group, genotype × fear interaction: $F_{(1,28)}$=56.73, p<0.0001; genotype effect:

*Figure 6 continued on next page*

*Figure 6 continued*

$F_{(1,28)}$=59.65, p<0.0001; fear effect: $F_{(1,28)}$=175.7, p<0.0001; two-way ANOVA with Bonferroni correction). (**C**) Representative image of the c-fos staining on the amygdala. Scale bar = 250 µm. Scale bar of the enlarged image = 50 µm. (**D**) Quantification of relative fluorescence intensity for c-fos signals, before and after fear conditioning on the amygdala (Control; WT and STAT3[PKO], Conditioned; WT and STAT3[PKO], respectively, n = 8 slices of three mice per experimental group, genotype × fear interaction: $F_{(1,28)}$=8.054, p=0.0084; genotype effect: $F_{(1,28)}$=8.601, p=0.0066; fear effect: $F_{(1,28)}$=124.7, p<0.0001; two-way ANOVA with Bonferroni correction). (**E**) Representative image of the c-fos staining on the prelimbic cortex. Scale bar = 250 µm. Scale bar of the enlarged image = 50 µm. (**F**) Quantification of relative fluorescence intensity for c-fos signals, before and after fear conditioning on the prelimbic cortex (Control; WT and STAT3[PKO], Conditioned; WT and STAT3[PKO], respectively, n = 8 slices of three mice per experimental group, genotype × fear interaction: $F_{(1,28)}$=20.15, p=0.0001; genotype effect: $F_{(1,28)}$=19.21, p=0.0001; fear effect: $F_{(1,28)}$=155.4, p<0.0001; two-way ANOVA with Bonferroni correction). Data are presented as mean ± SEM, and **p<0.01, ***p<0.001. ###p<0.001, compared with naïve WT group. The asterisk indicates the significance of difference between WT and STAT3[PKO] according to the presence or absence of fear conditioning, and the # sign indicates the significance of difference within the WT or STAT3[PKO] group according to the presence or absence of fear conditioning.

The online version of this article includes the following figure supplement(s) for figure 6:

**Figure supplement 1.** c-fos expression images and measurement on hippocampus.

**Figure supplement 2.** Three-dimensional reconstruction images of structural connections from Allen brain atlas database.

eye blink conditioning, the activity of PCs is reported to be suppressed during a conditioned stimulus after conditioning (***ten Brinke et al., 2015***). Although the conditioned response of PCs after fear conditioning has not been reported, the decreased output of PCs in STAT3[PKO] mice may disinhibit the activity of deep cerebellar nuclei (DCN), and DCN may send neural signals to other fear-related regions (***Figure 7***). Recently, it has been identified that the cerebellum has pathways to sensorimotor, associative, and modulatory forebrain (***Pisano et al., 2020***). As previous studies showed, fear memory network is distributed in the forebrain (***Wager et al., 2015***; ***Tovote et al., 2015***). This implicates that cerebello-cerebral connectivity may contribute to fear memory processing. In addition, cerebellar fastigial nucleus (FN)-ventrolateral periaqueductal grey (vlPAG) pathway is involved in the fear learning process (***Frontera et al., 2020***). It has been suggested that the suppression of DCN output facilitates freezing *via* reduction of activity in the vlPAG (***Vaaga et al., 2020***). The cerebellar outputs to the PAG negatively regulate freezing behavior. As our c-fos expression data suggested (***Figure 6***), the cerebellum may also positively regulate fear memory formation or retrieval *via* projecting to the other brain regions than the PAG such as thalamus either directly or indirectly. Therefore, the cerebellum may have distinct pathways to regulate freezing responses and fear memory. In the

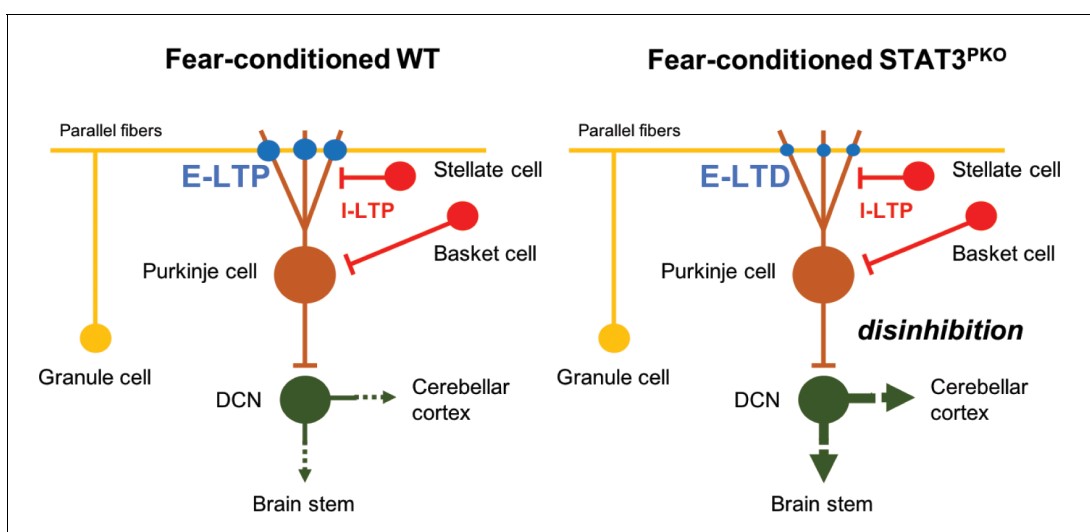

**Figure 7.** Model of current hypothesis: Purkinje cell (PC) output regulation in fear-conditioned wild-type (WT) and fear-conditioned STAT3[PKO] mice. In fear conditioning, long-term potentiation (LTP) occurs at both excitatory and inhibitory synapses (E: excitatory, I: inhibitory). In fear-conditioned STAT3[PKO] mice, reduced PC output led by long-term depression (LTD) at excitatory synapses disinhibits the activity of deep cerebellar nuclei (DCN) to the closed-loop circuitry of the cerebellum or to other fear-related regions while processing fear memory storage.

future, it may be necessary to identify how PCs regulate fear memory and fear-evoked freezing behavior through distinct connections.

It has been argued that long-term fear memory could be understood as a small world network (*Wheeler et al., 2013*; *Vetere et al., 2017*). Although previous researches excluded the cerebellum, there has been emerging evidence that the cerebellum-cortical/subcortical connections are involved in emotional processing (*Ernst et al., 2019*). Our behavioral results showed increased long-term fear memory in the STAT3[PKO] mice (*Figure 4C,D,E,F*). In tandem, neural activity in the paraventricular nucleus of the thalamus, basolateral amygdala, and prelimbic cortex increased in the fear-conditioned STAT3[PKO] mice, 24 hr after fear conditioning (*Figure 6*). These data imply that the whole brain fear network could be altered by the deletion of STAT3 in the PCs. Recent research suggests that inactivation of a single region disrupted the subcortical–cortical communication on global network organization (*Grayson et al., 2016*). In addition, we checked the possibility that physical connections exist among the fear-related areas. Guided by the Allen brain atlas database, we reconstructed the three-dimensional images for tracts between the cerebellum and the thalamus, the prelimbic cortex and the thalamus, and the basolateral amygdala and the thalamus (*Figure 6— figure supplement 2*). Although more experiments still remain to be conducted, these experimental data could be further analyzed using realistic computational models of the cerebellum–cortical/subcortical connectivity.

The cerebellum is progressively recognized for its cognitive role such as emotion, as shown in previous human brain imaging and clinical reports (*Stoodley and Schmahmann, 2009*; *Moberget et al., 2018*). Multiple animal studies also report that psychiatric disorders may result from cerebellar dysfunctions (*Tsai et al., 2012*). However, a well-characterized animal model for investigating the cerebellar role in fear memory is rare. PTSD-related genes and signaling molecules have long been predicted to act as memory enhancers or suppressors, particularly in the JAK2-STAT3 pathway (*Mynard et al., 2004*; *Hauger et al., 2012*; *Daskalakis et al., 2014*). Our findings suggest that STAT3 may act as a molecular switch to control fear behavior on the molecular and cellular basis of long-term fear memory storage. STAT3[PKO] mice showed the enhancement of long-term memory not only in Pavlovian fear conditioning but also in avoidance memories and fear-potentiated responses, which were less relevant to purely motor-related behaviors (*Figure 4*). Interestingly, STAT3[PKO] mice was shown to be normal on cerebellum-dependent motor learning. One possible explanation is that deletion of STAT3 in flocculus might not affect the induction of synaptic plasticity for motor learning. *Rest,* one of the target genes of *Stat3*, is expressed in vermis lobule V/VI, but rarely expressed in the flocculus (Allen brain atlas database). Lack of *Rest* expression may limit the effect of STAT3 deletion on the induction of synaptic plasticity in the flocculus and VOR learning. *Stat3*, and *Stat3*-targeting gene, *Hes-1*, are similarly expressed in the two cerebellar regions. Thus, the STAT3[PKO] mice can be proposed as the suitable genetic model to study cerebellar synaptic plasticity in relation to traumatic event-related psychiatric disorders.

In conclusion, we found that altered synaptic input strength in the STAT3[PKO] mice enhanced long-term fear memory. These results imply that STAT3 physiologically maintains the fear memory network by controlling the cerebellar synaptic plasticity, preventing aberrant fear behavior. This study furthers the evidence that output strength in the PCs modulates the neural activity of highly interconnected areas that consist of fear memory network.

## Materials and methods

**Key resources table**

| Reagent type (species) or resource | Designation | Source or reference | Identifiers | Additional information |
|---|---|---|---|---|
| Gene (*M. musculus*) | *Stat3* | GeneCards | | |
| Strain, strain background (*M. musculus, male*) | C57BL/6J | Jackson Laboratory | | |

*Continued on next page*

*Continued*

| Reagent type (species) or resource | Designation | Source or reference | Identifiers | Additional information |
|---|---|---|---|---|
| Genetic reagent (*M. musculus, male*) | Pcp2-Cre | Jackson Laboratory | #004146; RRID:IMSR_ JAX:004146 | Tg(Pcp2-cre) 2Mpin |
| Genetic reagent (*M. musculus, male*) | *Stat3*fl/fl | **Takeda et al., 1998** | | |
| Biological sample (*M. musculus*) | Primary hippocampal neurons | Jackson Laboratory | | Freshly isolated from *M. musculus* |
| Antibody | Anti-STAT3 (Rabbit, monoclonal) | Cell Signaling Technology | Cat. # 8768; RRID:AB_ 2722529 | IHC (1:200) |
| Antibody | Anti-STAT3 (Rabbit, monoclonal) | Cell Signaling Technology | Cat. # 4904; RRID:AB_331269 | WB: (1:1000) |
| Antibody | c-fos (Rabbit, monoclonal) | Cell Signaling Technology | Cat. # 2250; RRID:AB_ 2247211 | IF (1:200) |
| Antibody | Anti-GluA1 (Rabbit, polyclonal) | Abcam | Cat. # ab31232; RRID:AB_ 2113447 | WB (1:1000) |
| Antibody | Anti-CaMKII (Rabbit, monoclonal) | Abcam | Cat. # ab52476; RRID:AB_868641 | WB (1:1,000) |
| Antibody | Anti-PSD95 (Mouse, monoclonal) | Abcam | Cat. # ab2723; RRID:AB_303248 | WB (1:1000) |
| Antibody | Anti-GluA2 (Rabbit, polyclonal) | Synaptic Systems | 182 103; RRID:AB_ 2113732 | WB (1:1000) |
| Antibody | Anti-phospho-STAT3 (Rabbit, monoclonal) | Cell Signaling Technology | Cat. # 9145; RRID:AB_ 2491009 | WB (1:1000) |
| Antibody | Anti-α-tubulin (Mouse, monoclonal) | Santa Cruz | Cat. # sc8035; RRID:AB_628408 | WB (1:1000) |
| Antibody | Goat anti-mouse IgG | Enzo Life Science | Cat. # ADI-SAB-100-J; RRID:AB_ 11179634 | WB (1:10000) |
| Antibody | Goat anti-rabbit IgG | Enzo Life Science | Cat. # ADI-SAB-300-J; RRID:AB_ 11179983 | WB (1:10000) |
| Antibody | Cy3 donkey anti-rabbit IgG | BioLegend | Cat. # 406402; RRID:AB_893532 | IF (1:200) |
| Antibody | Biotinylated goat anti-rabbit IgG | Vector Laboratories | Cat. # BA-1000; RRID:AB_ 2313606 | IF (1:200) |
| Sequence-based reagent | *Stat3* | Qiagen | PCR primer | QT00148750 |
| Sequence-based reagent | *Hes-1* | Qiagen | PCR primer | QT00313537 |
| Sequence-based reagent | *Rest* | Qiagen | PCR primer | QT00116053 |
| Sequence-based reagent | *Gria1* | Qiagen | PCR primer | QT01062544 |
| Sequence-based reagent | *Gria2* | Qiagen | PCR primer | QT00140000 |

*Continued on next page*

*Continued*

| Reagent type (species) or resource | Designation | Source or reference | Identifiers | Additional information |
|---|---|---|---|---|
| Sequence-based reagent | *CamK2a_F* | This paper | PCR primer | ACGGAAGAGTA CCAGCTC TTCGAGG |
| Sequence-based reagent | *CamK2a_R* | This paper | PCR primer | CC TGGCCAGCC AGCACCTTCAC |
| Sequence-based reagent | *CamK2b_F* | This paper | PCR primer | GTCG TCCACAG AGACCTCAAG |
| Sequence-based reagent | *CamK2b_R* | This paper | PCR primer | CCAGATATCCA CTGGTTTGC |
| Commercial assay or kit | SMARTer Stranded Total RNA-Seq Kit v2-Pico Input | Takara | | |
| Chemical compound, drug | Oregon Green BAPTA 488 fluorescence dye | Molecular Probes | | |
| Chemical compound, drug | Stattic | Sigma-Aldrich | | |
| Software, algorithm | PatchMaster software | HEKA Elektronik | | |
| Software, algorithm | Mini Analysis Program | Synaptosoft | | |
| Software, algorithm | GraphPad Prism | GraphPad Software Inc | | |
| Software, algorithm | EthoVision XT 8.5 | Noldus | | |

## Experimental model and subject details

Mice carrying a Cre transgene under the control of the Pcp2 promoter (Pcp2-Cre$^{+/+}$) were purchased from the Jackson Laboratory (#004146, Tg(Pcp2-cre)2Mpin, Bar Harbor, ME, USA). STAT3 floxed (*Stat3*$^{fl/fl}$) mice were kindly gifted from Dr. S Akira (Osaka University, Japan) (*Kwon et al., 2017*; *Takeda et al., 1998*). Mice with a STAT3 deletion in PCs were generated by crossing mice with the floxed STAT3 allele with mice expressing Cre under the control of the Pcp2 promoter. The genetic backgrounds for both the Cre and floxed STAT lines were C57BL/6J. Genotyping was performed as previously described (*Kwon et al., 2017*). The primers were specific for exons 22 and 23 of *Stat3*. All experiments were performed with male mice aged 8–10 weeks. Experimental animals were maintained under specific pathogen-free conditions and 22 ± 1°C with a reversed 12 hr light–dark cycle (lights on at 07:00 hr). All experimental procedures were reviewed and approved by the Institutional Animal Care and Use Committee at the College of Medicine, Seoul National University.

## Immunohistochemistry

Immunohistochemical staining was performed as previously described (*Kwon et al., 2017*). In brief, brains of mice were perfused with buffer containing 4% paraformaldehyde, and each brain region (cerebellum, thalamus, amygdala, prelimbic cortex, and hippocampus) embedded in paraffin. The paraffin blocks were cut using a microtome (4 μm, Finesse E+, Thermo Shandon, Runcom, UK). Paraffin slices were mounted on the silane-coated micro slides (Muto Pure Chemicals, Tokyo, Japan) and then allowed to air dry. Before immunostaining, the slides were deparaffinized in xylene, dehydrated through graded alcohols, and heated in citrate buffer. Nonspecific binding was blocked with 5% normal goat serum (Vector Laboratories, Burlingame, CA) in PBS. The immunostaining was performed with primary antibodies for STAT3 (#8768, Cell Signaling Technology, MA, USA) and visualized using biotinylated goat anti-rabbit IgG (BA-1000, Vector Laboratories, CA, USA). Signals were developed with the Vectastain ABC kit (PK-4001, Vector Laboratories, CA, USA), DAB reagents

(K5007, Dako, CA, USA), and counterstained with hematoxylin (S3309, Dako, CA, USA). The slides were dehydrated through graded alcohols and mounted with Vectashield mounting medium (H-1000, Vector Laboratories, CA, USA). Images were obtained using the Leopard program on a microscope (BX53; Olympus, Tokyo, Japan). For immunofluorescence, brain sections (thalamus, amygdala, and prelimbic cortex) were stained with the primary antibodies for c-fos (9F6) (#2250, Cell Signaling, MA, USA) and visualized using Cy3 donkey anti-rabbit IgG (#406402, BioLegend, CA). The slides were mounted with 4',6-diamidino-2- phenylindole (DAPI), and images were collected using the LSM510 program on a confocal microscope (Carl Zeiss MicroImaging, München, Germany). The c-fos protein levels were quantified using ImageJ by measuring the integrated density of the c-fos fluorescence intensity, and the relative fluorescence intensity was measured by calculating the relative value of the integrated density of the WT control.

## Cellular imaging and dendritic arborization analysis

Each PC dendrite was three-dimensionally imaged with Oregon Green BAPTA 488 fluorescence dye (Molecular Probes, O6807, USA) by using a whole cell recording method. Then, 50–100 slice images were merged onto a single image. Dendrites were traced with the aid of Zen software (Carl Zeiss MicroImaging, München, Germany) for quantitative analysis using ImageJ. Branch lines were defined by a branch point and the branch points or dendritic tips were terminated by each of the daughter segments. All lengths were measured manually. The number of dendritic spines was also counted from the same images.

## Laser capture microdissection

Laser capture microdissection (LCM, Molecular Machines and Industries) was used to isolate the group of cells from cerebellar slice tissues, with the aid of a laser beam under direct microscopic visualization. PC layers were isolated directly by cutting target regions away from unwanted cell layers. The obtained cell populations were used to generate cDNA libraries and to analyze RNA-seq transcriptome. Tissue was frozen on specialized coated membrane slides.

## RNA sequencing analysis

RNA was isolated using NucleoSpin RNA XS (Macherey-Nagel, Germany), and the sequencing library was prepared with SMARTer Stranded Total RNA-Seq Kit v2-Pico Input (Takara, Shiga, Japan) and sequenced with Illumina HiSeq 2500. Fastq was aligned to GRCm38 using STAR-2.4.2b, and estimated read count and TPM were calculated with RSEM-1.3.0. DEG was calculated by DESeq2 with expected read count generated from RSEM. Low expressed genes with total read count less than one among all samples were filtered. DEG was defined as Benjamini–Hochberg corrected p-value $\leq 0.05$, absolute expression difference over 1.5-fold, and baseMean value $\geq 1$. Upregulated and downregulated genes were queried for pathway analysis with g: Profiler. Normalization was performed with DESeq2 via regularized log (rlog) transformation method. Values were gene/sample centered and normalized. Unsupervised hierarchical clustering was performed with Cluster 3.0. Heatmap was plotted with Java Treeview.

## Slice preparation and electrophysiology

Mice were anesthetized with isoflurane and decapitated, and the brains were immediately removed and placed in ice-cold slicing solution (0–4°C) containing the following artificial cerebrospinal fluid: 124 mM NaCl, 2.5 mM KCl, 1 mM NaH$_2$PO$_4$, 1.3 mM MgCl$_2$, 2.5 mM CaCl$_2$, 26.2 mM NaHCO$_3$, and 20 mM D-glucose, bubbled with a gas mixture of 5% CO$_2$/95% O$_2$ to maintain a pH of 7.4. Sagittal slices of the cerebellar vermis (250–300 μm thick) were obtained using a Vibratome (Leica VT1200S; Leica, Nussloch, Germany). For recovery, slices were incubated at 28°C for 30 min. All recordings were performed within 6–8 hr from recovery. Whole-cell recordings in the cerebellar PCs were performed in the voltage/current-clamp mode using an amplifier (HEKA Instruments, Lambrecht/Pfalz, Germany). The signal was low-pass filtered at 5 kHz, and acquired at 10 kHz. For recording the miniature excitatory postsynaptic currents (mEPSCs), the recording electrodes (resistance 2–4 MΩ) were filled with a solution containing 135 mM Cs-methanesulfonate, 10 mM CsCl, 10 mM HEPES, 4 mM Mg$_2$ATP, 0.4 mM Na$_3$GTP, and 0.2 mM EGTA (pH 7.25). For miniature IPSC (mIPSC) recording, 135 mM Cs-methanesulfonate was replaced by 140 mM CsCl. We used recording pipettes (3–4 MΩ)

filled with the following: 9 mM KCl, 10 mM KOH, 120 mM K-gluconate, 3.48 mM MgCl$_2$, 10 mM HEPES, 4 mM NaCl, 4 mM Na$_2$ATP, 0.4 mM Na$_3$GTP, and 17.5 mM sucrose, pH 7.25, for testing synaptic plasticity. Data were acquired using an EPC8 patch-clamp amplifier (HEKA Elektronik) and PatchMaster software (HEKA Elektronik). All electrophysiological traces were acquired in lobule V–VI of cerebellar vermis. Synaptic responses were analyzed by Mini Analysis Program, Synaptosoft.

## Western blot

Western blotting was performed as previously described (*Kwon et al., 2017*). The nitrocellulose membranes were probed with the primary antibodies for GluA1(ab31232, abcam, Cambridge, UK), CaMKII(ab52476, abcam, Cambridge, UK), PSD95 (ab2723, abcam, Cambridge, UK), GluA2 (#182 103, Synaptic Systems, Göttingen, Germany), phospho-STAT3, STAT3 (#9145, #4904, Cell Signaling Technology, MA, USA), and α-tubulin (sc-8035, Santa Cruz Biotechnology, CA, USA) for the target molecules, followed by HRP-conjugated secondary antibodies for goat anti-mouse IgG, and goat anti-rabbit IgG (Enzo Life Science, NY, USA). The membranes were visualized using an ECL detection kit (SurModics, MN, USA).

## Primary hippocampal neuron culture

Primary hippocampal neurons were isolated from P1 C57BL/6 mice by dissociating with 0.25% trypsin and plated onto poly-L-lysine (Sigma-Aldrich, MO, USA)-coated culture dish. Primary neurons were grown in neurobasal medium (Gibco, CA, USA) containing B27 (Gibco, CA, USA), 2 mM Gluta-MAX-I supplement (Gibco, CA, USA), and 100 μg/ml penicillin/streptomycin (Gibco, CA, USA), and incubated at 37°C in a humidified condition of 95% O$_2$/5% CO$_2$. Primary neurons were seeded onto 6-well culture dishes coated with poly-L-lysine, and treated with 10 μM (in 0.1% DMSO) stattic (Sigma-Aldrich, MO, USA) for 24 hr.

## Quantitative real-time PCR

Total RNA was isolated from brain tissues and cell lysates using an RNAiso Plus reagent (Takara, Shiga, Japan) and cDNA was synthesized using ReverTra Ace qPCR RT Master Mix (Toyobo, Osaka, Japan). Quantitative real-time PCR was performed using the EvaGreen qPCR Mastermix (Applied Biological Materials, BC, Canada), and the results were normalized to the signals of GAPDH expression. Primers for *Stat3* (QT00148750), *Hes-1* (QT00313537), *Rest* (QT00116053), *Gria1* (QT01062544), and *Gria2* (QT00140000) were purchased from Qiagen (Germantown, MD, USA). *CamK2a* was amplified using PCR primers: 5′-ACGGAAGAGTACCAGCTCTTCGAGG-3′ and 5′-CCTGGCCAGCCAGCACCTTCAC-3′. *CamK2b* was amplified using PCR primers: 5′-GTCGTCCACAGAGACCTCAAG-3′ and 5′-CCAGATATCCACTGGTTTGC-3′.

## Fear conditioning test

WT and STAT3[PKO] mice (males, at least 8 weeks old) were trained in a basic Skinner box module (Mouse Test Cage), and underwent fear memory acquisition. After 3 min of free exploration in the conditioning chamber, a series of conditioned stimuli (3000 Hz tones amplified to 85–90 dB lasting 30 s) was administered five times at 30 s intervals. The last 1 s of each CS was paired with the US consisting of an electric foot shock (0.7 mA). The mice were left there for an additional few minutes and were returned to their home cages.

Fear memory retrieval was tested at 10 min and 24 hr after the acquisition session in the independent groups of mice. The subjects were placed inside the conditioning box and left there for 3 min. After the acquisition session, we tested short-term memory fear retention by presenting to the mice the same context in which they were trained (contextual fear conditioning). A few minutes later, the subjects were placed in a novel environment (cylindrical container) for an additional 3 min, and two acoustic stimuli (CS) were administered, identical to those used during the acquisition session (cued fear conditioning). Long-term memory was tested in the same way 24 hr after the training session. In all experiments, the freezing response was recorded and its duration was taken as a fear index. Freezing was defined as the complete absence of somatic motility, except for respiratory movements. All animals were used exclusively for the fear conditioning test.

## Avoidance test

During the acquisition session, animals were placed in the light compartment of the apparatus (Gemini, San Diego Ins.). When the animal innately crossed to the dark compartment, it received one-foot shock (1 mA for 1 s). During the retention test, each animal was placed in the light compartment, and a few seconds later a guillotine door was opened, allowing them to enter the dark compartment. The latency crossing into the dark compartment was recorded. The test session finished either when the animals went into the dark compartment or remained in the light compartment for 300 s. During the test session, no electric shock was applied. All animals were used exclusively for the avoidance test.

## Acoustic startle response and prepulse inhibition

For testing, animals were placed into the startle apparatus (SR-LAB-Startle Response System, San Diego Instruments, USA) and allowed a 5 min acclimation period. The startle session started with three successive startle stimuli of 30 ms duration (75–120 dB). Within each block, individual trials were randomly distributed. Four different intensities of acoustic prepulse stimuli (80, 90, 100, and 120 dB) were used, each prepulse being 30 ms in duration. Movement of the animal within the cylinder was measured by a piezoelectric accelerometer. The representation of the acoustic stimulus and piezoelectric response of the accelerometer was controlled and digitized by the SR-LAB software and interface system. In all experiments, the variable interval between trials averaged 10 s; hence, each session lasted approximately 15 min. All animals were used exclusively for the startle response and prepulse inhibition test.

## Open field test

The open field consisted of $40 \times 40$ cm polyvinyl chloride square with 40 cm walls. Mice were placed in the center of an open field box, and their movements were recorded with a video camera for 30 min. The total distance traveled and time spent in the central zone ($20 \times 20$ cm) were calculated using video tracking software (EthoVision XT 8.5, Noldus). All animals were used exclusively for the open field test.

## Elevated plus maze test

The elevated plus maze consisted of two open arms ($30 \times 5$ cm) and two closed arms ($30 \times 5 \times 15$ cm) connected by a central square ($5 \times 5$ cm). The whole maze was raised 50 cm above the ground. Mice were placed in the central square of the maze facing the open arms. The movements of the mice were recorded during a 5 min test period. The number of entries and time spent in the open and closed arms were calculated using video tracking software (EthoVision XT 8.5, Noldus). All animals were used exclusively for the elevated plus maze test.

## Rotarod test

The rotarod test was performed by a coordination test system (Rotamex 5, Columbus). All animals were pre-trained (for 5 min) on the rotarod in order to reach a stable performance and the rotarod test was performed for five consecutive days. The mice were placed on a rotating rod (3 cm in diameter) that accelerated from 3 to 50 rpm for 6 min and the latency to fall was recorded. All animals were used exclusively for the rotarod test.

## VOR test

Two basal ocular–motor responses, which are VOR in dark (dVOR), and VOR in light (lVOR), were measured. For dVOR and lVOR, turntable stimulation was applied in sinusoidal rotation with ±5° of rotation amplitude. The dVOR and lVOR were conducted under light off and on conditions, respectively. Each response was recorded at four different rotating frequencies (0.1, 0.25, 0.5, and 1.0 Hz). All animals were used exclusively for the VOR test.

## Quantification and statistical analysis

An appropriate sample size was computed when the study was being designed. Before we started the statistical tests, such as two-tailed Student's $t$-test, and ANOVA, we confirmed that data we obtained were passed the normality test. If not, we carried out the Mann–Whitney test. Two

independent group comparisons of immunohistochemical staining, RT-PCR, western blotting, electrophysiological experiments, and behavioral experiments such as contextual and cued fear memory, avoidance, open field, and elevated plus maze tests were analyzed by two-tailed Student's *t*-test or Mann–Whitney test. Two-way ANOVA was applied to evaluate behavioral experiments such as fear acquisition, acoustic startle response, rotarod, and VOR tests, and to analyze synaptic transmission, plasticity, and intrinsic excitability with time or stimulus strength or injected currents. Bonferroni's *post hoc* test was performed if applicable. All data are presented as mean ± SEM. All statistical analyses were performed using GraphPad Prism software (GraphPad Software Inc).

## Acknowledgements

This study was supported by grants from the NRF funded by the Korea government (MISP; 2014R1A2A1A11053203 to S-K.Y. and NRF-2018R1A5A2025964 to S.J.K.); S-H.K. received a scholarship from the BK21-plus education program provided by the National Research Foundation of Korea (NRF). This manuscript was grammatically edited by Jae-Rong Ahn (Tufts University).

## Additional information

### Funding

| Funder | Grant reference number | Author |
| --- | --- | --- |
| National Research Foundation of Korea | NRF-2018R1A5A2025964 | Sang Jeong Kim |
| Ministry of Health and Welfare | 2014R1A2A1A11053203 | Sang-Kyu Ye |

The funders had no role in study design, data collection and interpretation, or the decision to submit the work for publication.

### Author contributions

Jeong-Kyu Han, Conceptualization, Resources, Data curation, Formal analysis, Investigation, Visualization, Methodology, Writing - original draft, Project administration, Writing - review and editing; Sun-Ho Kwon, Conceptualization, Resources, Data curation, Formal analysis, Investigation, Visualization, Methodology, Project administration, Writing - review and editing; Yong Gyu Kim, Resources, Software, Methodology; Jaeyong Choi, Data curation, Software, Methodology; Jong-Il Kim, Sang-Kyu Ye, Supervision, Funding acquisition; Yong-Seok Lee, Supervision, Writing - review and editing; Sang Jeong Kim, Conceptualization, Supervision, Funding acquisition, Writing - review and editing

### Author ORCIDs

Yong Gyu Kim http://orcid.org/0000-0002-4378-6248
Jong-Il Kim http://orcid.org/0000-0002-7240-3744
Yong-Seok Lee http://orcid.org/0000-0002-6217-9574
Sang Jeong Kim https://orcid.org/0000-0001-8931-3713

### Ethics

Animal experimentation: This study was reviewed and approved by the Institutional Animal Care and Use Committee at the College of Medicine, Seoul National University. Experimental animals were maintained under specific pathogen-free conditions and 22±1°C with a reversed 12 h light-dark cycle (Permit Number: 200122-6). All animal sacrifice was performed under iso- propanol anesthesia, and every effort was made to minimize suffering.

### Decision letter and Author response

Decision letter https://doi.org/10.7554/eLife.63291.sa1
Author response https://doi.org/10.7554/eLife.63291.sa2

## Additional files

### Supplementary files

• Supplementary file 1. List of up/down genes from analysis of Kyoto Encyclopedia of Genes and Genomes (KEGG) pathway has been applied to RNA-seq differential expression analyses.

• Transparent reporting form

### Data availability

All data generated or analysed during this study are included in the manuscript and supporting files.

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
