## [Decision Letter]

**Acceptance summary:**

This intriguing and rigorous set of experiments indicate a role of cerebellar plasticity in long-term memory of cued fear conditioning. The results add to the growing body of evidence for a contribution of the cerebellum to functions previously ascribed to the forebrain, and provide major new insights about cellular and circuit level mechanisms for this contribution. These studies are needed to understand the contribution of the cerebellum to emotional control, which has been studied primarily in humans.

**Decision letter after peer review:**

Thank you for submitting your article "Ablation of STAT3 in Purkinje Cells Reorganizes Cerebellar Synaptic Plasticity in Long-Term Fear Memory Network" for consideration by *eLife*. Your article has been reviewed by three peer reviewers, and the evaluation has been overseen by Jennifer Raymond as the Reviewing Editor and Gary Westbrook as the Senior Editor. The following individuals involved in review of your submission have agreed to reveal their identity: Dagmar Timmann (Reviewer #2); Keiko Tanaka-Yamamoto (Reviewer #3).

The reviewers have discussed the reviews with one another and the Reviewing Editor has drafted this decision to help you prepare a revised submission.

Summary:

This is an intriguing and rigorous set of experiments suggesting that STAT3 in cerebellar Purkinje cells play a critical role in long-term memory of cued fear conditioning. The authors show that ablation of STAT3 in Purkinje cells leads to genomic changes, such as upregulating pathways important for synaptic signaling. There is also enhanced AMPA-R expression, increased basal excitatory synaptic transmission, and impaired LTP of the parallel fiber-Purkinje cell synapses. These changes in cerebellar function are associated with enhanced long-term memory of cued-fear conditioning as well as enhanced neural activity in brain regions that are thought to be important for cued conditioning--the thalamus, amygdala and prelimbic cortex. Interestingly, this enhancement of fear learning is specific to cued conditioning as there were no differences in contextual fear conditioning. The results add to the growing body of evidence for a contribution of the cerebellum to functions previously ascribed to the forebrain, and provide major new insights about cellular and circuit level mechanisms for this contribution. These studies are needed to understand the contribution of the cerebellum to emotional control, which has been studied primarily in humans

Essential revisions:

The specificity of the effects of manipulating STAT3 in the Purkinje cells on cued fear conditioning and not contextual fear conditioning or motor learning tasks such as the rotorod or VOR learning is striking and surprising. On the one hand, the selective effect on cued fear conditioning, and not contextual fear conditioning helps to rule out an indirect effect of motor deficits on the freezing behavioral readout of fear conditioning. On the other hand, the reviewers wondered whether STAT3/cerebellar LTP plays a role in other forms of aversive conditioning, and why there was no effect of STAT3 knockout on motor performance or motor learning, given the well-established contribution of the cerebellum to those functions. At the very least, the authors should discuss how the specificity might arise, addressing the specific points below. Additional data that provide insights about the specificity would be welcomed, but were not deemed necessary.

– Why was there no effect on contextual conditioning? If the authors have results of c-fos in hippocampus (presumably no activity increase?) and could show them, this would help to explain the specific effects on cued memory, but not on contextual memory.

– Is STAT3 in Purkinge cells expected to alter other types of aversive learning?

– LTD and LTP at PF-PC synapses have been implicated in motor learning. Considering a current theory of VOR learning that gain-up learning relies on LTD and gain-down learning relies on LTP (e.g. Titley and Hansel, Cerebellum, 2016), one might have expected abnormal gain-down learning in STAT3-PKO mice, because of impaired LTP. However, VOR learning was shown to be normal in STAT3-PKO mice (Figure 4M). To avoid confusion, some reasonable explanations would be necessary. One possibility is that, unlike lobe V/VI, the induction of synaptic plasticity would not be affected by deletion of STAT3 in flocculus, a cerebellar lobe responsible for VOR learning. Is there a difference in expression of STAT3 or other molecules having similar functions between cerebellar regions?

– The rotarod test may not have been sensitive enough to detect motor deficits. Even wild-type mice could stay on the rod for about 80 seconds out of 360 seconds after 5 days of learning. Might the testing conditions be too difficult to detect any difference in motor coordination between wild-type and STAT3-PKO mice?

---

## [Author Response]

Essential revisions:The specificity of the effects of manipulating STAT3 in the Purkinje cells on cued fear conditioning and not contextual fear conditioning or motor learning tasks such as the rotorod or VOR learning is striking and surprising. On the one hand, the selective effect on cued fear conditioning, and not contextual fear conditioning helps to rule out an indirect effect of motor deficits on the freezing behavioral readout of fear conditioning. On the other hand, the reviewers wondered whether STAT3/cerebellar LTP plays a role in other forms of aversive conditioning, and why there was no effect of STAT3 knockout on motor performance or motor learning, given the well-established contribution of the cerebellum to those functions. At the very least, the authors should discuss how the specificity might arise, addressing the specific points below. Additional data that provide insights about the specificity would be welcomed, but were not deemed necessary.– Why was there no effect on contextual conditioning? If the authors have results of c-fos in hippocampus (presumably no activity increase?) and could show them, this would help to explain the specific effects on cued memory, but not on contextual memory.

We appreciate reviewer’s comment. Followed by this comment, c-fos expression was checked in hippocampus. The c-fos expressions were increased in both WT and STAT3-PKO mice after tone stimulation, while we did not find any difference between WT and STAT3-PKO. We added the result in Figure 6—figure supplement 1, and described in the last part of the Results section. This result suggests that altered cerebellar synaptic plasticity may not affect hippocampal-dependent fear memory.

– Is STAT3 in Purkinge cells expected to alter other types of aversive learning?

As the reviewer commented, STAT3-PKO mice showed enhanced learning in passive avoidance test and enhanced fear potentiated startle response, suggesting that STAT3 deletion in Purkinje cells affected other types of aversive learning in addition to classical fear conditioning. These results are shown in Figure 4D and F.

– LTD and LTP at PF-PC synapses have been implicated in motor learning. Considering a current theory of VOR learning that gain-up learning relies on LTD and gain-down learning relies on LTP (e.g. Titley and Hansel, Cerebellum, 2016), one might have expected abnormal gain-down learning in STAT3-PKO mice, because of impaired LTP. However, VOR learning was shown to be normal in STAT3-PKO mice (Figure 4M). To avoid confusion, some reasonable explanations would be necessary. One possibility is that, unlike lobe V/VI, the induction of synaptic plasticity would not be affected by deletion of STAT3 in flocculus, a cerebellar lobe responsible for VOR learning. Is there a difference in expression of STAT3 or other molecules having similar functions between cerebellar regions?

As the reviewer commented, *Rest*, one of the target genes of *Stat3*, is expressed in lobule V/VI, but rarely expressed in the flocculus (Allen brain atlas database). Lack of *Rest* expression may limit the effect of STAT3 deletion on the induction of synaptic plasticity in the flocculus and VOR learning. *Stat3*, and *Stat3 –* targeting gene, *Hes-1*, are similarly expressed in the two cerebellar regions. This explanation is described in the sixth paragraph of the Discussion section of the revised manuscript.

– The rotarod test may not have been sensitive enough to detect motor deficits. Even wild-type mice could stay on the rod for about 80 seconds out of 360 seconds after 5 days of learning. Might the testing conditions be too difficult to detect any difference in motor coordination between wild-type and STAT3-PKO mice?

For the rotarod test, an accelerating rotarod was used. The apparatus was accelerated from 3~4 to 40~50 rpm. This is a commonly used protocol in numerous previous studies using mutant mice. The latency to fall was measured with a cutoff time of 2~6 min. This method allows us to reveal subtle phenotype in motor learning, because it tests several sessions for learning curve. Our own data is based on the same protocol described above. And we found no significant difference in locomotion test between wild-type and STAT3-PKO mice, as well as VOR learning. All things considered, we would like to address that the testing conditions above are likely to make difference in motor learning and coordination.